**COMMUNICATIONS**

# Spinning-enabled wireless amphibious origami millirobot

Qiji Ze[1,4], Shuai Wu[1,4], Jize Dai[1], Sophie Leanza[1], Gentaro Ikeda [2], Phillip C. Yang[2], Gianluca Iaccarino[1,3] & Ruike Renee Zhao [1✉]

Wireless millimeter-scale origami robots have recently been explored with great potential for biomedical applications. Existing millimeter-scale origami devices usually require separate geometrical components for locomotion and functions. Additionally, none of them can achieve both on-ground and in-water locomotion. Here we report a magnetically actuated amphibious origami millirobot that integrates capabilities of spinning-enabled multimodal locomotion, delivery of liquid medicine, and cargo transportation with wireless operation. This millirobot takes full advantage of the geometrical features and folding/unfolding capability of Kresling origami, a triangulated hollow cylinder, to fulfill multifunction: its geometrical features are exploited for generating omnidirectional locomotion in various working environments through rolling, flipping, and spinning-induced propulsion; the folding/unfolding is utilized as a pumping mechanism for controlled delivery of liquid medicine; furthermore, the spinning motion provides a sucking mechanism for targeted solid cargo transportation. We anticipate the amphibious origami millirobots can potentially serve as minimally invasive devices for biomedical diagnoses and treatments.

[1] Department of Mechanical Engineering, Stanford University, Stanford, CA 94305, USA. [2] Stanford Cardiovascular Institute and Division of Cardiovascular Medicine, Department of Medicine, Stanford University School of Medicine, Stanford, CA 94305, USA. [3] Institute for Computational and Mathematical Engineering, Stanford University, Stanford, CA 94305, USA. [4] These authors contributed equally: Qiji Ze, Shuai Wu. ✉email: rrzhao@stanford.edu

Millimeter-scale origami robots[1–3] that can be operated wirelessly to locomote in narrow spaces and morph their shapes for specific tasks have recently been explored with great potential for biomedical applications, such as disease diagnosis[4,5], targeted drug delivery[6,7], and minimally invasive surgery[8,9]. Existing millimeter-scale origami devices[10,11] usually require separate geometrical components for locomotion and functions, which increase the complexity of the robotic systems and their operation[12,13]. In addition, most origami robots exhibit limited locomotion modes[14–16], and none of them can achieve both on-ground and in-water locomotion. This makes it challenging for existing robots to adaptively navigate complex, unstructured ground, and aqueous environments, such as those often seen in biomedical environments.

Rotation-based locomotion, including rolling[17,18], flipping[19,20], and spinning[21,22], is widely used by robotic systems to generate fast on-ground or in-water translational motion due to their high speed and efficiency. Compared to other locomotion mechanisms such as crawling, wriggling, and walking that are commonly found in small-scale flexible robotic systems[23–26], the rolling of highly symmetric structures[27,28] like spheres and geodesic polyhedrons provides effective and fast on-ground locomotion (Fig. 1a). In addition, a high level of structural symmetry enables smoother steering for omnidirectional motion compared to less symmetric systems, which usually require extra steps to change their moving direction[29,30]. Rotation also generates the most effective in-water propulsion, implemented by the engineered propeller whose blades extend radially and rotate to exert linear thrust in water (Fig. 1a). Although rotational motions effectively facilitate both on-ground and in-water locomotion, it requires differently designed structures due to the distinct motion mechanisms. For small-scale applications such as in biomedical fields, a remotely actuated amphibious miniature robot that can better exploit and integrate rotation-enabled on-ground and in-water locomotion is highly beneficial, especially in confined hybrid environments, such as in the urinary system and gastrointestinal system.

In this work, we report a rotation-enabled amphibious origami millirobot that integrates capabilities of multimodal locomotion, delivery of liquid medicine, and cargo transportation with wireless operation. This origami millirobot breaks the conventional way of utilizing origami folding only for shape reconfiguration and integrates multiple functions in one simple body. The geometrical features of the origami are exploited for generating omnidirectional locomotion in various working environments, including on the unstructured ground, in liquids, and at air–liquid interfaces through rolling, flipping, and spinning-induced propulsion. The folding/unfolding capability of the origami is utilized as a pumping mechanism for the controlled delivery of liquid medicine. In addition, the spinning motion provides a sucking mechanism for targeted solid cargo transportation. The capabilities of multimodal locomotion and integrated functions could enable the millirobot as a potential minimally invasive device for biomedical diagnoses and treatments.

## Results

### Magnetically actuated amphibious origami millirobot.

The amphibious millirobot is designed based on the Kresling origami[31,32], a triangulated hollow cylinder. Figure 1b shows an example of such millirobots with a cross-section diameter of 7.8 mm (see Supplementary Fig. 1 for fabrication process). Globally, the Kresling pattern possesses high geometrical symmetry with its aspect ratio ($H/D$) close to 1, making it resemble a Kresling "sphere" that allows on-ground rolling and flipping

around the axes parallel and perpendicular to its longitudinal direction, respectively (Fig. 1c). Locally, the Kresling has tilted triangular panels, which function in a way that is analogous to propeller blades. As shown in Fig. 1c, when the Kresling spins about its longitudinal axis, the tilted triangular panels can effectively generate propulsion for in-water swimming.

To bring forth the rolling, flipping, and spinning locomotion modes, magnetic actuation[33–36] is adopted to remotely manipulate the motions of the Kresling millirobot. This separates the power source and control system from the robot, enabling miniaturized machines for small-scale biomedical applications. As shown in Fig. 1d, the Kresling millirobot is prepared by attaching a thin magnetic plate to one hexagonal end of the Kresling. The magnetic plate has a volume $V$ and an in-plane magnetization $\mathbf{M}$ (see Supplementary Fig. 2 for detailed information of the magnetic plate). By applying a magnetic field $\mathbf{B}$, the magnetic plate generates a torque regulated by $\mathbf{T} = V(\mathbf{M} \times \mathbf{B})$, which results in the robot's rigid body rotation to align its magnetization with the magnetic field. In the presence of a continuously rotating magnetic field (see Supplementary Fig. 3 for the magnetic actuation setup), the robot's magnetization follows the magnetic field, leading to the continuous rotation for rolling, flipping, or spinning. As shown in Fig. 1e–g, the locomotion mode of the robot depends on its rotational axis and its interaction with the working environment. When on the ground, the robot demonstrates rolling or flipping modes (Fig. 1c). This is achieved by rotating the magnetic field in the plane that is perpendicular (Fig. 1e) or parallel (Fig. 1f) to the longitudinal axis of the robot, respectively. While in water, the robot swims (Fig. 1g) by spinning about its longitudinal axis.

Besides the different locomotion modes along a straight line, the robot can easily navigate upon manipulating its orientation under the three-dimensional magnetic field without adding extra mechanism designs or actuators[37,38], which is also beneficial for small-scale applications. Figure 1h shows the steering mechanism of the robot. Since the robot's magnetization $\mathbf{M}$ always follows the applied rotating magnetic field $\mathbf{B}$ in its plane of rotation, altering the rotation axis of $\mathbf{B}$ would effectively change the robot's orientation by its rigid body rotation.

In addition, the Kresling is a foldable shell whose internal cavity and its twist-induced contraction[39] permit the robot's integrated multifunctionality, such as the pumping mechanism induced by folding/unfolding (Fig. 1i) for controlled release of liquid medicine. With two magnetic plates attached on the two ends of the robot as shown in Fig. 1i, applying an instant magnetic field would generate a pair of magnetic torques $+T$ and $-T$ to fold the robot. We also design the Kresling to be monostable (see Supplementary Figs. 4 and 5 for mechanical characterization), so it recovers to its unfolded state autonomously once the magnetic field is removed, forming a pumping mechanism enabled by cyclic folding/unfolding actuation. The mechanisms and performances of all locomotion modes and integrated functions will be discussed in detail in the following sections.

### Self-adaptive locomotion on unstructured ground with controlled delivery of liquid medicine.

In addition to locomotion on flat ground, the adaptive locomotion on unstructured ground with different terrains is crucial to promoting robotic applications in different environments. Most existing robots need to actively switch their configuration[40,41] or control strategy[42,43] when facing different surface features such as ridges, stairs, and potholes. Here, we demonstrate that the self-adaptive locomotion of the Kresling robot can navigate on different terrains by overcoming various obstacles through a self-selected locomotion

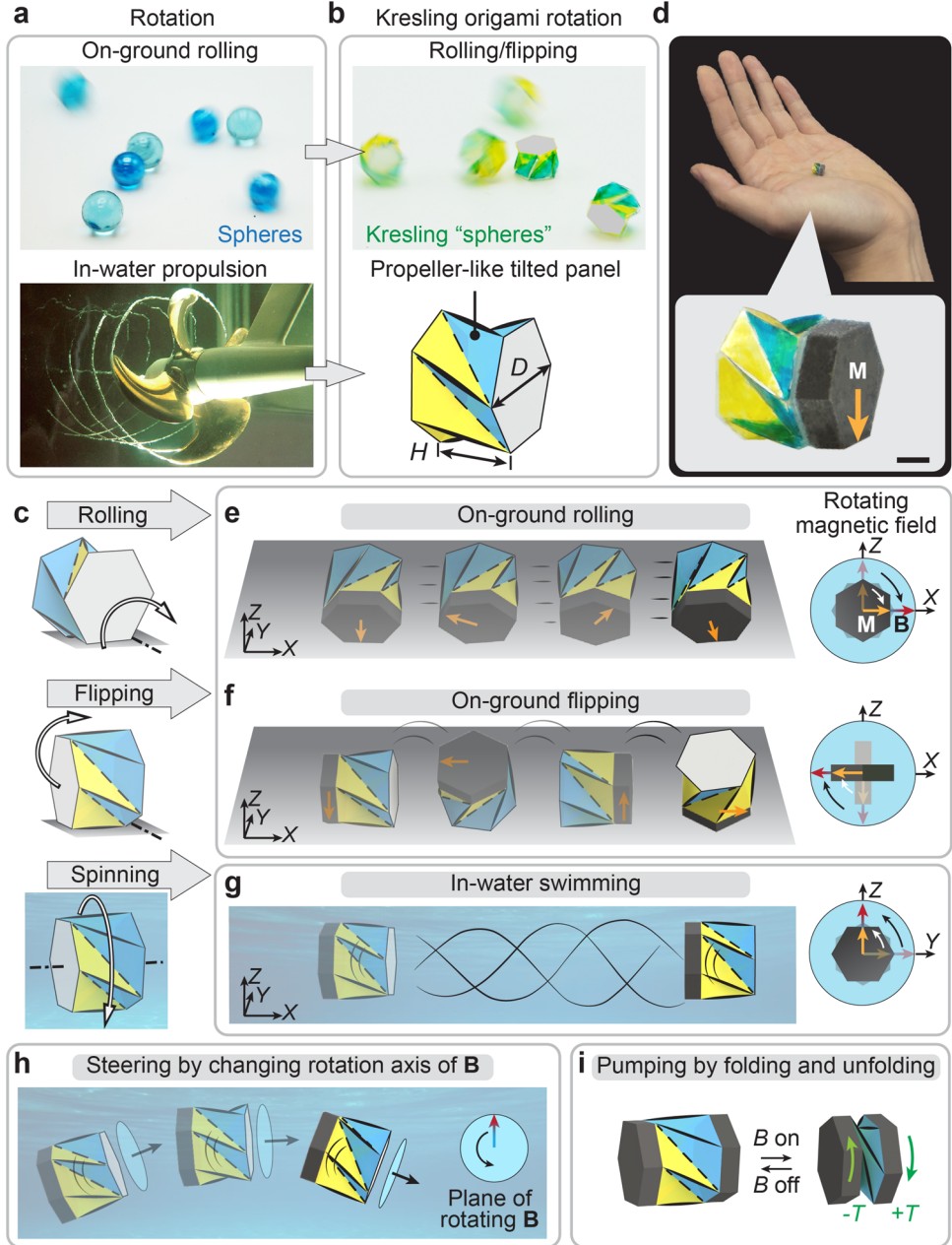

**Fig. 1 Mechanisms of the magnetically actuated amphibious origami robot. a** Rotational motions for on-ground omnidirectional rolling of spheres and in-water propulsion of propeller. The propeller image was reproduced with permission from Rolls-Royce. **b** Kresling "spheres" with high global symmetry for omnidirectional motion and locally tilted triangular panels for in-water propulsion. **c** Three locomotion modes, rolling, flipping, and spinning, based on the robot's rotational motions around different axes (dashed lines) in different environments. **d** Image of the robot composed of a Kresling with a magnetic plate attached to it. The magnetic plate has an in-plane magnetization **M**. Scale bar: 2 mm. Mechanisms of (**e**) on-ground rolling, (**f**) on-ground flipping, and (**g**) in-water swimming actuated by a rotating magnetic field **B**. **h** Steering mechanism by changing rotation axis of **B**. **i** Pumping mechanism by cyclic folding/unfolding upon a pair of magnetic torques +T and -T.

mode based on the surface features. Compared to those robots whose moving direction requires a specific orientation of the robot realized by complex control, our Kresling robot shows agile navigation capability as its moving direction is independent of the robot's orientation. When a moving direction for the robot is specified, despite the random initial orientation of the robot, the rotating magnetic field would always move the robot toward the desired direction (Supplementary Movie 1). When the robot interacts with obstacles or rough surfaces, it can automatically switch between rolling and flipping (Fig. 2a) to adapt to the terrain features while maintaining its moving direction without

the need of changing the controlling magnetic field. As shown in Fig. 2b, the robot's longitudinal axis is parallel to the surface initially. A rotating **B** ($B = 10$ mT, $f = 4$ Hz. See Supplementary Fig. 6 for the magnetic field profile) in the $XZ$-plane rolls the robot clockwise along the positive $X$-direction, where $Z$ is the outward normal of the surface. When the robot's rolling is perturbed by an obstacle, such as a high wall, it automatically switches to the flipping mode and overcomes the obstacle without adjusting the rotating magnetic field. After that, it returns to rolling again toward the predetermined direction (Supplementary Movie 1). Figure 2c and Supplementary Fig. 7 demonstrate the

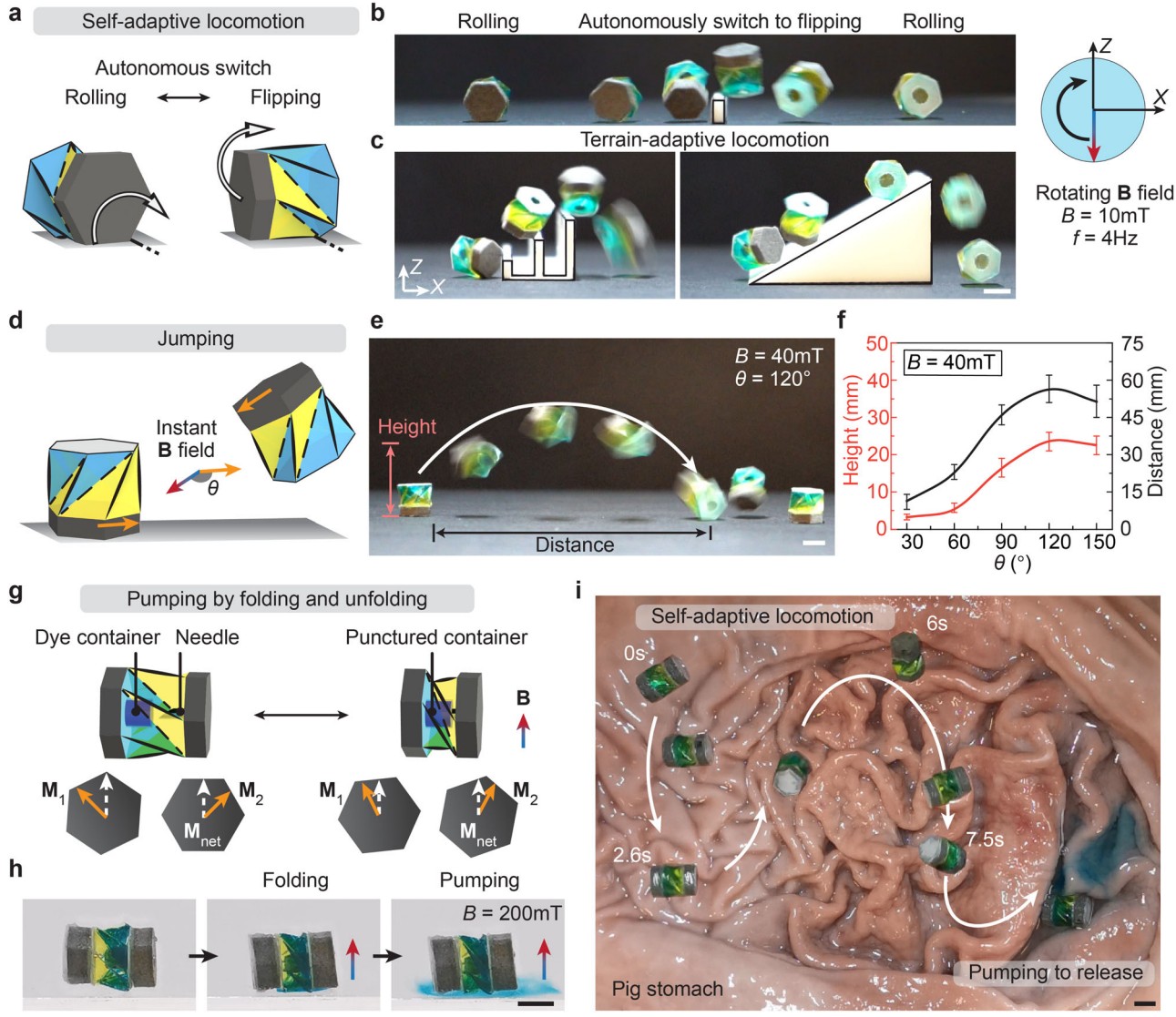

**Fig. 2 Self-adaptive on-ground locomotion and controlled delivery of liquid medicine. a** Mechanism of self-adaptive locomotion. Under the same rotating magnetic field with a frequency (*f*) of 4 Hz and a strength (*B*) of 10 mT, the robot overcomes different terrains, including (**b**) a wall obstacle, (**c**) spaced stairs, and an inclined surface by autonomously switching between rolling and flipping. **d** Mechanism of jumping enabled by an instant magnetic field with a strength of *B* and an angle of *θ* between magnetic field and magnetization. **e** Demonstration of jumping when *B* = 40 mT and *θ* = 120°. **f** Characterization of jumping performance. **g** Controlled release of liquid medicine by the magnetically actuated pumping mechanism. **h** Experimental images of pumping medicine. The two attached magnetic plates have the same dimensions but different in-plane magnetization directions **M**₁ and **M**₂. **M**_net is the net magnetization of the Kresling robot. **i** Controlled delivery of liquid medicine by self-adaptive locomotion and pumping in an ex vivo pig stomach. Scale bars: 5 mm.

robot moving on different terrains with various obstacles about the same size as the robot, including spaced stairs (4 mm gap and 4 mm increasing height), an inclined surface (30° slope angle), a column array (4 mm height and 8 mm distance), and a cylinder obstacle (4 mm diameter), showing the robustness of the self-adaptive locomotion (Supplementary Movie 2). The capability to autonomously choose the locomotion modes to go over various terrains along the designated direction (positive *X* axis) significantly reduces the control complexity. This is particularly beneficial for navigating on complex unpredictable terrain features and also helps reduce the size of the robots.

For larger obstacles that are no longer easily overcome by on-ground flipping and rolling, the robot can jump over them upon applying an instant magnetic field, as shown in Fig. 2d. The jumping height and direction are determined by the magnetic field strength *B* and the relative angle *θ* between the magnetic field and the magnetization of the magnetic plate. Under a 40 mT

instant magnetic field and 120° relative angle, the robot can achieve a jumping height of 23.5 mm and a distance of 56.2 mm (Fig. 2e and Supplementary Movie 3). The jumping height and distance with respect to the relative angle *θ* are characterized for *B* = 40 mT, as shown in Fig. 2f (Supplementary Movie 3).

The Kresling robot's rigid body locomotion can be further coupled with its folding capability, which is enabled by its shell structure, for multifunctionality with integrated navigation and controlled release of liquid medicine. The internal cavity and the magnetically actuated pumping mechanism of the Kresling are utilized for storage and on-demand release of liquid medicine. To achieve this, two magnetic plates are attached to the two hexagonal ends of the Kresling robot, with a needle and a dye container attached to the two inner ends, as shown in Fig. 2g (see Supplementary Fig. 8 for the sample fabrication). The two magnetic plates have the same dimensions but different in-plane magnetization directions (**M**₁ and **M**₂, note that |*M*₁| = |*M*₂|),

resulting in a net magnetization $\mathbf{M}_{net}$ which is the vector sum of $\mathbf{M}_1$ and $\mathbf{M}_2$. When an external magnetic field $\mathbf{B}$ is applied along the direction of $\mathbf{M}_{net}$, magnetic torques $+T$ and $-T$ regulated by $|\mathbf{T}| = V|\mathbf{M}_1 \times \mathbf{B}| = V|\mathbf{M}_2 \times \mathbf{B}|$ are generated to rotate the magnetic plates and fold the Kresling. Upon the removal of the magnetic field, the monostable structure recovers to its unfolded state, forming a reversible folding/unfolding actuation as the pumping mechanism. Here, the angle between $\mathbf{M}_1$ and $\mathbf{M}_2$ is 90° (see Supplementary Figs. 9–12 for the criteria of angle selection based on folding efficiency experiments and analytical calculations). As illustrated in Fig. 2h, when applying a 200 mT magnetic field $\mathbf{B}$, the induced magnetic torques contract the Kresling, during which the needle punctures the dye container to release the "medicine" (Supplementary Movie 4). At the same time, the Kresling's internal cavity shrinks and squeezes the "medicine" out through the radial cuts. By repeating the folding/unfolding actuation cyclically, the "medicine" is gradually pumped out with a controllable dose.

To test the self-adaptive locomotion and the integrated multifunctionality in the biomedical environment, we conduct the experiment in an ex vivo pig stomach with rugae and mucosa, as shown in Fig. 2i (Supplementary Movie 4). Following a predetermined path, the robot successfully demonstrates effective self-adaptive locomotion on the compliant, viscous, and unstructured stomach surface through the combination of rolling and flipping. After reaching the targeted location, the robot releases the controlled dose of "medicine" by magnetic pumping.

**Spinning-enabled swimming under water and at air–water interface**. Amphibians adapt to different environments, including on ground, in water, and through transitional zones. Motivated by this, amphibious robots[44–46] with high adaptivity have drawn great attention for broad applications, such as resource exploration, disaster rescue, and military reconnaissance. However, these amphibious robots usually need to combine different structures, actuation systems, or control strategies to realize locomotion both on the ground and in water. Taking advantage of the Kresling's geometric features to interact with water for propulsion, we demonstrate the robot's swimming both underwater and at air–water interfaces by its magnetically actuated spinning motion. The Kresling structure possesses tilted triangular panels distributed radially along its axis, which function in a way that is analogous to the blades of a propeller (Fig. 3a). As shown in Fig. 3b, the Kresling has the tilted panels' orientation similar to the right-handed propeller (the thumb points to the moving direction while the four fingers curl to the rotation direction of the propeller). When applying a rotating magnetic field about the Kresling's axis at a frequency of $f$, the robot spins. Following the right-hand rule, when the robot spins about the predetermined moving direction in Fig. 3b, the blue panels push back the water, resulting in a net propulsion force for forward swimming. In addition, we modify the Kresling robot (see Supplementary Fig. 1 for the modified Kresling pattern) to have one frontal hole and six radial cuts serving as the major inlet and outlets of water flow, respectively (Fig. 3c). This fluid dynamic interaction within and outside the robot results in better swimming performance. The measured swimming speed in Fig. 3d reveals the effective propelling performance of the Kresling structure. The swimming speed increases with the rotating frequency of the magnetic field. In addition, the robot with the hole and cuts demonstrates a much faster speed with a maximum value of 81.2 mm s$^{-1}$ (11.9 body length s$^{-1}$) under a rotating magnetic field with $B = 10$ mT and $f = 30$ Hz, while the maximum speed of the robot without the hole and cuts is 66.0 mm s$^{-1}$ (9.7 body length s$^{-1}$) under the same magnetic field.

To qualitatively understand the mechanism of enhanced swimming efficiency for the robot with the hole and cuts, computational fluid dynamics (CFD) simulations are conducted to evaluate two rigid robot geometries with and without the hole and cuts during robot spinning. The simulations are performed by imposing a rotational speed and external flow velocity corresponding to the robot spinning and swimming speeds in the experiments. As illustrated in Fig. 3e, for the robot without the hole and cut, the gauge pressure along the robot's rotation axis increases at the stagnation point due to flow impingement, leading to a high swimming resistance. In contrast, for the robot with the hole and cuts, the hole enables the robot to capture fluid that is then centrifuged out through the cuts, together with a significantly decreased pressure in the front of the robot due to the absence of the stagnation point. The frontal pressure drop leads to a reduced swimming resistance, which corresponds to an increased swimming speed.

Figure 3f shows the horizontal swimming of the robot in a water tank, with rheoscopic fluid added to visualize the turbulent flow (Supplementary Movie 5). Note that the density of the robot is adjusted to be close to that of water to minimize the influence of gravity (see the section of "Sample Fabrication" in Supplementary text for detailed information). Benefiting from the convenient navigation enabled by magnetic actuation, Fig. 3g, h demonstrate the agile swimming performance of the robot along a 2D "∞" path and a 3D spiral path, respectively, by continuously changing the rotation axis of the applied magnetic field (Supplementary Movie 6). The swimming locomotion of the robot can also be integrated with the pumping mechanism to design multifunctional robots for navigation and targeted delivery of liquid medicine (Supplementary Movie 7, see Supplementary Fig. 13 for detailed information). A large-scale version of the Kresling robot is also fabricated to better illustrate the contraction-based pumping mechanism for a multitarget release process of liquid medicine (Supplementary Movie 8 and Supplementary Fig. 14).

Besides swimming underwater, the robot can also navigate at air–water interfaces by the same swimming mechanism with the help of a self-capturing air bubble to actively tune its effective density. As shown in Fig. 3i (Supplementary Movie 9), the robot swims from the bottom of the tank at a 45° pitching angle. When approaching the interface, the frontal hole of the robot pops out of the water to trap an air bubble with an increasing size, which gradually lowers the robot density for swimming and navigation at the air–water interface. Utilizing the captured air bubble and surface tension, the robot can float on the water after removing the magnetic field. Note that we distinguish the outer and inner surfaces of the Kresling to be hydrophobic and hydrophilic to allow for easy water penetration into the robot's interior through the frontal and radial cuts for controllable robot density (see Supplementary text for the information of surface treatment). To let the robot sink back into the water, a magnetic field perpendicular to the water surface can be applied to change the robot's orientation, disengaging the Kresling's outer hydrophobic hexagonal surface from the air–water interface and allowing water to enter the robot interior to push out the air bubble.

**Amphibious locomotion across different environments with cargo transportation**. The amphibious robot is capable of navigating across different environments, including on ground, in liquid, and through transitional zones for multifunctional operations. We demonstrate that the robot can transport cargo in a hybrid terrestrial-aquatic environment (Fig. 4a) by utilizing different spinning-enabled locomotion and a spinning-enabled sucking mechanism. As shown in Fig. 4b, upon swimming to the

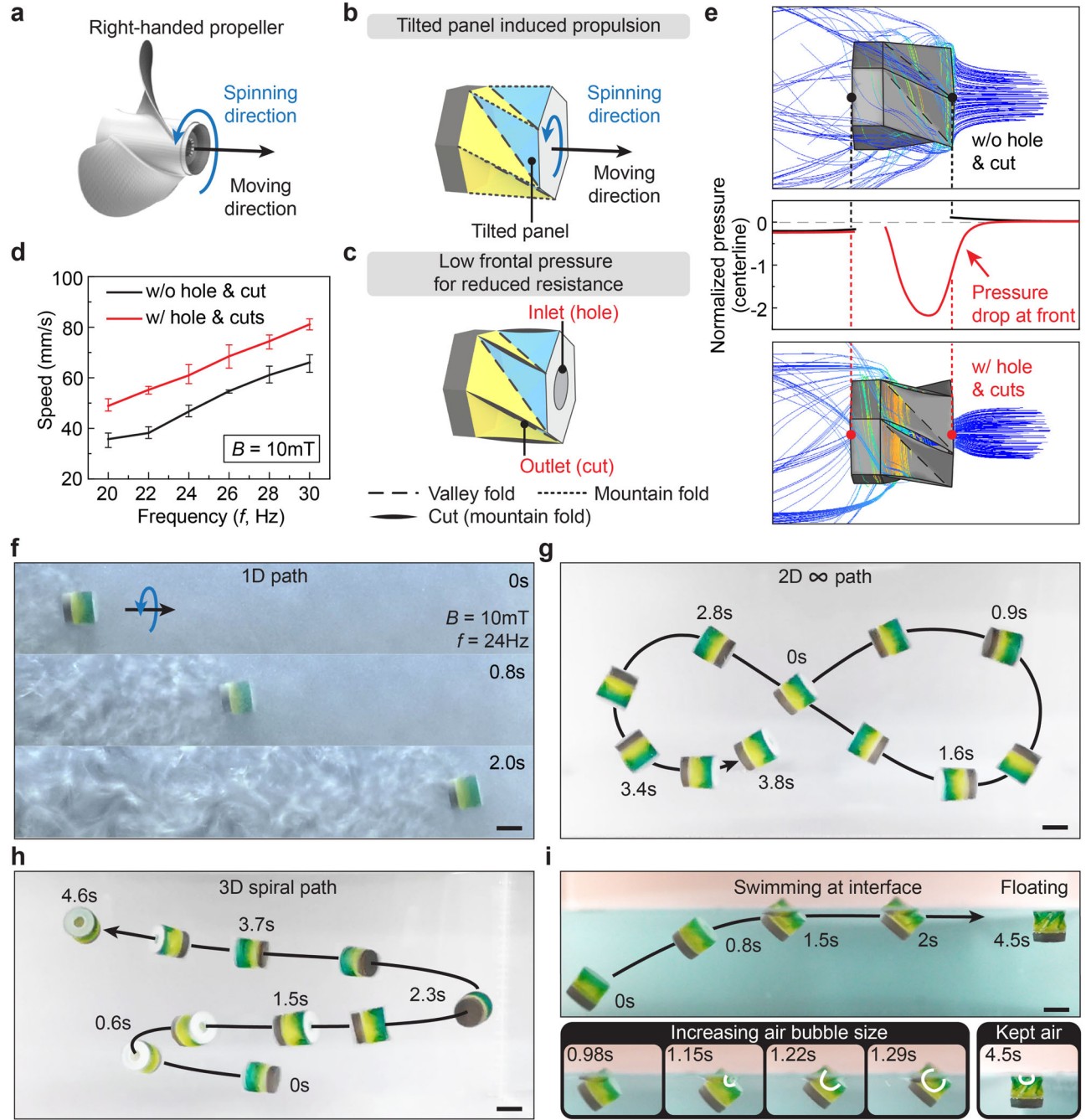

**Fig. 3 Swimming mechanisms and navigation underwater and at air–water interface. a** A right-handed propeller. **b** Propulsion induced from propeller-like tilted panels of the Kresling. **c** Modified Kresling with frontal hole and radial cuts for enhanced swimming performance. **d** Speed comparison of the horizontal swimming for robots with and without the hole and cuts under different magnetic field rotation frequencies. **e** CFD simulations for comparison of the streamlines and normalized pressures of the robots with and without the hole and cuts. Demonstrations of swimming under a rotating magnetic field with a strength of 10 mT and a frequency of 24 Hz along (**f**) a straight line, (**g**) a 2D "∞" path, and (**h**) a 3D spiral path. **i** Swimming at the air–water interface. Scale bars: 5 mm.

front of the cargo, the spinning robot generates a low pressure at the hole region that sucks the cargo through the hole (Supplementary Movie 10). Once the robot reaches the target position, the cargo can be released by its gravity when the hole of the robot is facing the ground. In the hybrid terrestrial-aquatic environment (Supplementary Movie 11), the robot initially moves over different on-ground terrains by self-adaptive locomotion through automatically switching between rolling and flipping based on the terrain features (Fig. 4c). The robot jumps over a high barrier to a shallow water area and then rolls and submerges into the water

(Fig. 4d). After capturing the cargo (Fig. 4d), the robot swims to the target underwater ground and releases it (Fig. 4e). Finally, the robot returns to the initial location through underwater swimming, air–water interface swimming, followed by self-adaptive locomotion over continuous stairs (Fig. 4f).

An ex vivo pig stomach filled with viscous fluid is used to illustrate the amphibious locomotion of the robot in the biomedical environment (Supplementary Movie 12). The viscosity of the fluid (12 mPa S measured at a shear rate of $22\ s^{-1}$) is in the range of the viscosity of gastric juice[47,48]. By manipulating the

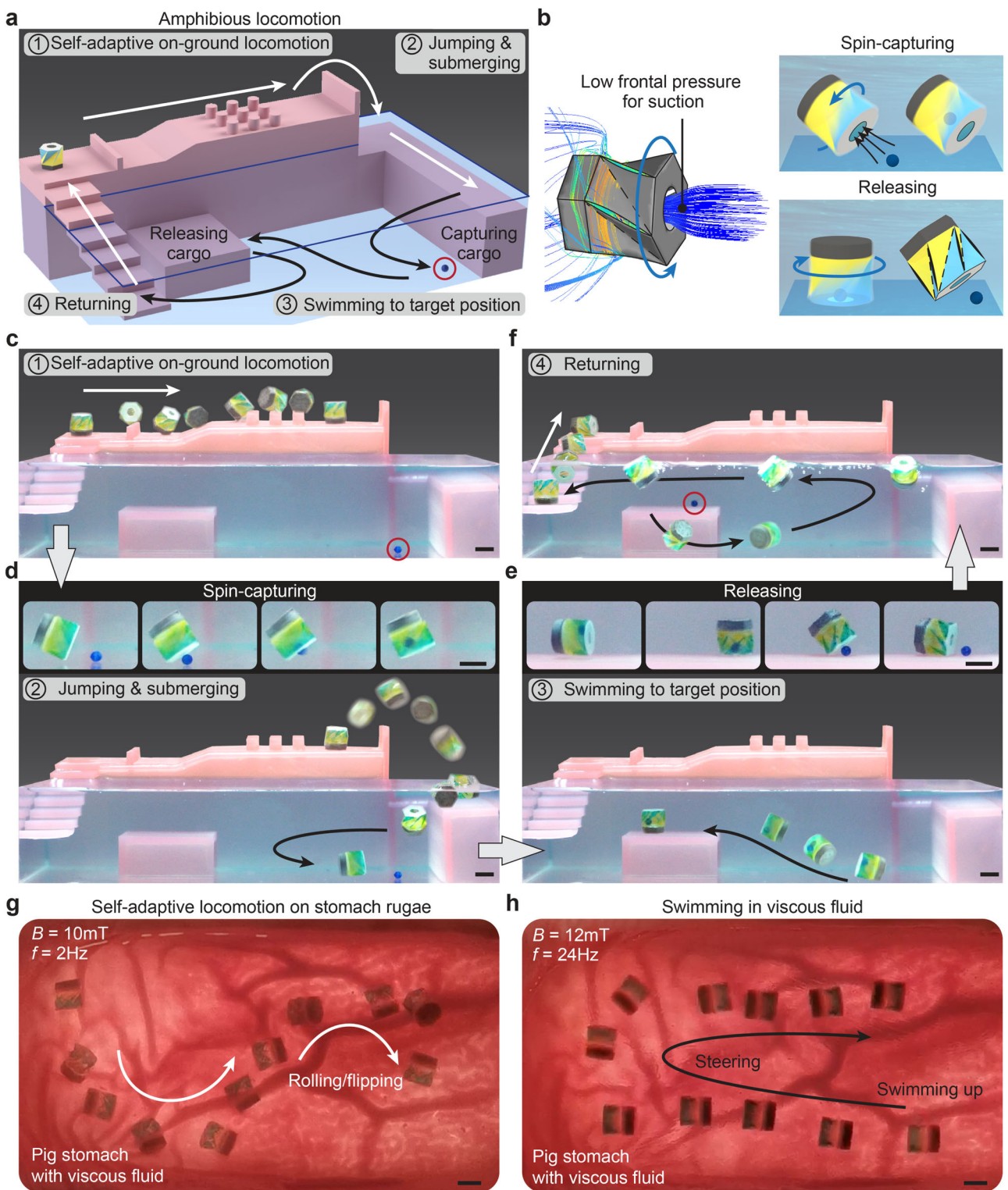

**Fig. 4 Amphibious locomotion in hybrid terrestrial-aquatic environments and targeted cargo transportation. a** The environment model for the demonstration of amphibious locomotion and cargo transportation. **b** Schematics of cargo capturing via a spinning-enable sucking mechanism and cargo releasing. **c** The robot rolls and flips over different terrains in a self-adaptive manner. **d** The robot jumps over a barrier to a shallow water area, submerges into a deep water area, and swims toward the cargo to capture it. **e** The robot swims to the targeted position and releases the cargo. **f** The robot swims to the air–water interface and returns to the initial position over continuous stairs by self-adaptive locomotion. The demonstration of (**g**) self-adaptive on-ground rolling/flipping and (**h**) swimming in an ex vivo pig stomach with viscous fluid (Viscosity: 12 mPa S). Scale bars: 5 mm.

applied rotating magnetic field, the robot demonstrates effective and continuous self-adaptive on-ground rolling and flipping on the rugged stomach surface (Fig. 4g) and swimming in the viscous fluid (Fig. 4h). Compared to water, the viscous fluid induces a larger resistance, therefore, the magnetic field strength ($B = 12$ mT) used here for swimming is slightly larger than it is used in demonstrations in water ($B = 10$ mT) to provide a larger driving force. In-depth research on the influence of the fluid viscosity on the swimming performance of our robot can be conducted in the future.

## Discussion

Here, we have demonstrated a wireless amphibious origami millirobot based on the triangulated cylindrical Kresling origami structure and magnetically actuated rotational motions. By taking full advantage of the interaction between geometric features of the Kresling and external working environments, the robot has achieved multiple rotation-enabled locomotion modes in an adaptive manner, including rolling, flipping, and swimming, in various terrains across ground and liquid. In addition, the robot has exhibited integrated multifunctional applications permitted by its foldable thin-shell structure, including controlled delivery of liquid medicine using the pumping mechanism induced by reversible folding/unfolding and targeted cargo transportation using the spinning-enabled sucking mechanism. Experiments in the ex vivo animal organs reveal the potential applications of our robot in complex biomedical environments, such as the gastro-intestinal tract. One step further, the proposed concept of the spinning-enabled amphibious origami robot can be scaled up or down for broader applications. With advanced fabrication methods[49], downsized origami robots can be manufactured for biomedical environments including blood vessels and ureters. In addition, the internal cavity of Kresling can be utilized to integrate different components within the robot, such as mini cameras and forceps, enabling multiple biomedical operations, including endoscopy and biopsy. We anticipate that the multifunctional magnetic amphibious origami millirobots could facilitate a wide range of future minimally invasive devices for biomedical diagnoses and treatments with better functionality and less damage to patients.

## Methods

**Fabrication of the Kresling Millirobot**. The amphibious origami millirobots are fabricated by assembling Kresling origami and one magnetic plate (Supplementary Fig. 2), except for the millirobot for controlled release of liquid medicine, which needs two magnetic plates for the folding capability (Supplementary Fig. 8). The Kresling sample is folded from the designed two-dimensional flower-shaped pattern (Supplementary Fig. 1a), which is cut from 0.05 mm thick polypropylene film. Before cutting the Kresling pattern, one side of the polypropylene film is treated with hydrophilic coating (Hydrophilic Coating 8-3C, Coatings2Go LLC, USA) to distinguish the outer and inner surfaces of the Kresling to be hydrophobic and hydrophilic to enable the easy water penetration into the interior of the robot. After the pattern is folded, 0.127 mm thick Mylar hexagons with and without a 3-mm hole are attached to the pattern's top and bottom sides, respectively, to produce the Kresling with one frontal hole and six radial cuts for multifunctionality. The magnetic plates are made of Ecoflex-0030 silicone (Smooth-On, Inc., USA) embedded with 10 vol% hard-magnetic particles (NdFeB, average size of 100 μm, Magnequench, Singapore) and 20 vol% glass bubbles (K20 series, 3M, USA).

**Magnetic actuation**. The locomotion of the Kresling robot is controlled under a uniform three-dimensional magnetic field within a space of 160 mm by 120 mm by 80 mm, which is generated by customized 3D Helmholtz coils (Supplementary Fig. 3). A cylinder N52 neodymium permanent magnet with a diameter of 50 mm and a thickness of 25 mm is used to fold the Kresling for controlled release of liquid medicine (Supplementary Fig. 9).

**Computational fluid dynamics simulations**. The commercial software Ansys Fluent (ANSYS, Inc., USA) is utilized for computational fluid dynamics (CFD) simulations to qualitatively study the differences induced by the presence of the robot's frontal hole and radial cuts.

More details about sample fabrication, material characterization, and experimental setup are provided in the Supplementary Materials.

## Data availability

All data generated in this study are provided in the main text or the Supplementary Materials.

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

## Acknowledgements

R.R.Z., Q.Z., S.W., J.D., and S.L. acknowledge support from NSF Career Award CMMI-2145601 and NSF Award CMMI-2142789. G. Ikeda acknowledges support from the American Heart Association 20POST35120540. R.R.Z acknowledges Prof. Kyung-Suk Kim of Brown University for useful discussions on the fluid dynamics of the robotic swimmer.

## Author contributions

R.R.Z. designed research and supervised the study. Q.Z., S.W., and R.R.Z. designed the robot and all experiments. Q.Z., S.W., J.D., and S.L. fabricated the samples, performed the experiments, and analyzed experimental data. G. Ikeda, and P.C.Y. did the surgery on the pig to extract the stomach. Q.Z. and S.W. performed the experiments in pig stomach. G. Iaccarino performed the CFD simulations and analyzed numerical results. Q.Z., S.W., S.L., G. Iaccarino, and R.R.Z. wrote the manuscript. All authors reviewed the manuscript.

## Competing interests

A provisional patent has been filed.
