## [Peer Review File · Nature Communications]

Spinning-enabled Wireless Amphibious Origami MillirobotReviewers' Comments:

Reviewer #1:

Remarks to the Author:

The paper by Zhao and her team presents an amazing work on using magnetic material and origami to design innovative robots that are millimeter in size. Both magnetic material-based soft robots and origami robots are research areas that are intensively pursued. However, most magnetic material-based SoRos take very simple form and most origami robots use folding to change the shape for either morphing or locomotion, making these robots have relatively simple function. To achieve more sophisticated functions, most research uses a linear addition approach, i.e. adding a sensor if navigation is needed, adding an origami mechanism if shape morphing is needed, etc. This often results in bulky systems with many single-function units. In this paper, the authors cleverly used the geometry of Kresling origami for locomotion, origami folding for functions, and magnetic materials for remote control. This tight integration of different features of materials and structures allows them to build very small robots yet capable to carry out very sophisticated functions. This certainly is one of the best robots this reviewer has ever seen over years. The paper is also nicely written and enjoyable to read. I highly recommend the publication of this paper in Nature Communication (or even maybe in Nature). I have the following minor questions.

1. For spinning induced swimming, is there a speed limit on how fast the robot can swim? This reviewer feels that the spinner may lose propulsion if it spins too fast.
2. Also for spinning induced swimming, how fast the robot can respond to a change of direction in the magnetic field? What are the controlling factors in determining the response speed?
3. Can the authors comment on the feasibility of extending their approach to other form of origami? Or maybe Kresling is the only one due to its unique shape and folding mechanism?
4. For the Kresling pattern used, the cross-section has a hexagonal shape. Can other shape of cross-section be used (for example, pentagon shape)?

Reviewer #2:

Remarks to the Author:

This article reported a wireless multi-modal origami millirobot capable of on-ground rolling/flipping, obstacle crossing, jumping, spinning-enabled swimming under water and at the interface, with integrated functions, such as medicine delivery and cargo transportation. In this work, the Kresling origami structure is incorporated into the millirobot. Based on the geometrical features of Kresling origami, the self-adaptive rolling-to-flipping on-ground omnidirectional movement can be achieved, when the robot crosses an obstacle (e.g., wall, spaced stairs, and inclined surface). When two magnetic plates are integrated into the robot, the compressive and rotational deformations of Kresling structure can be achieved to enable the release of medicine. Taking advantage of the Kresling's tilted triangular panels distributed radially along its axis, a net propulsion force can be generated when the robot is actuated by magnetic field spins. The demonstrations that the amphibious robot can move in various man-made environments and ex vivo pig stomach with/without viscous fluid were further provided. Overall, the millirobot reported in this paper is very impressive, with a rich diversity of locomotion modes. The paper is thereby recommended for publication in Nature Communications, after minor revisions noted below.

Comments:

1. In Fig. 2C, the author shows that the robot can move on an inclined surface of 30°. What is the maximum angle of the inclined surface on which the millirobot could climb?
2. In Fig. 3F-I, the swimming performances of the robots along different routes are demonstrated. When the plane of rotated magnetic field is parallel to the axis of Kresling structure in the beginning, can the robot adaptively adjust its swimming mode and eventually move in a spinning mode?
3. On Page 3, Line 92 of supplementary information, the Kresling structure can be assembled into

robot after experiencing 400-time cyclic compressions, for consistent and monostable mechanical behavior. Is it possible to achieve desired mechanical property of the Kresling structure without cyclic compressions?

4. Please check the typo in Movie S5. 'Titled' should be corrected as 'tilted'.

Responses to Reviewers

Reviewer #1

General Comments:

The paper by Zhao and her team presents an amazing work on using magnetic material and origami to design innovative robots that are millimeter in size. Both magnetic material-based soft robots and origami robots are research areas that are intensively pursued. However, most magnetic material-based SoRos take very simple form and most origami robots use folding to change the shape for either morphing or locomotion, making these robots have relatively simple function. To achieve more sophisticated functions, most research uses a linear addition approach, i.e. adding a sensor if navigation is needed, adding an origami mechanism if shape morphing is needed, etc. This often results in bulky systems with many single-function units. In this paper, the authors cleverly used the geometry of Kresling origami for locomotion, origami folding for functions, and magnetic materials for remote control. This tight integration of different features of materials and structures allows them to build very small robots yet capable to carry out very sophisticated functions. This certainly is one of the best robots this reviewer has ever seen over years. The paper is also nicely written and enjoyable to read.

I highly recommend the publication of this paper in Nature Communication (or even maybe in Nature). I have the following minor questions.

Response: We do appreciate the reviewer's positive comments about our work and manuscript. Following, we have provided point-by-point addressments of the specific comments from the reviewer.

Specific Comments:

Comment #1 - For spinning induced swimming, is there a speed limit on how fast the robot can swim? This reviewer feels that the spinner may lose propulsion if it spins too fast.

Response: We would like to thank the reviewer for bringing up this good question. Within the frequency range we tested (20-30 Hz), the higher the magnetic spinning frequency is, the higher the swimming speed. The spinning frequency we used in this work already provides us with a reasonably high swimming speed. In general, if the magnetic torque induced by the applied field is large enough for the robot to overcome the water resistance while spinning, the propulsion and resultant swimming speed increase with the spinning frequency. When the spinning frequency is larger than a specific value (usually referred as step-out frequency ^[R1, R2]), the magnetic torque cannot sustain the spinner's rotation anymore. The spinner will lose stability, and its speed will drop to zero immediately. The step-out frequency and corresponding maximum swimming speed can be enhanced by increasing the applied magnetic torque, which is regulated by the applied magnetic field strength (B), the magnetization (M_r) and the volume (V) of the magnetic materials on the spinner.

In our paper, all the above three parameters are fixed ($B=10$ mT, $M_r=58.21$ kA m⁻¹, and $V= 46.27$ mm³). The step-out frequency and corresponding maximum speed are about 30 Hz and 81.2 mm s⁻¹. If we increase B , M_r , or V , the maximum speed of the spinner can be further enhanced.

It is worth mentioning that the B of 12 mT is used for the demonstration in viscous fluid while the B of 10 mT is used for other demonstrations in water. This is due to the increased swimming resistance from the viscous fluid compared to water. The magnetic torque under B of 10 mT cannot sustain the spinner's rotation at the frequency of 24 Hz.

Based on the reviewer's comment, we have revised the following contents in the revised main text.

“Compared to water, the viscous fluid induces a larger resistance, therefore, the magnetic field strength ($B = 12$ mT) used here for swimming is slightly larger than it is used in demonstrations in water ($B = 10$ mT) to provide larger driving force.”

[R1] Peyer, K. E., Zhang, L., & Nelson, B. J. (2013). Bio-inspired magnetic swimming microrobots for biomedical applications. *Nanoscale*, 5(4), 1259-1272.

[R2] Mahoney, A. W., Nelson, N. D., Peyer, K. E., Nelson, B. J., & Abbott, J. J. (2014). Behavior of rotating magnetic microrobots above the step-out frequency with application to control of multi-microrobot systems. *Applied Physics Letters*, 104(14), 144101.

Comment #2 - Also for spinning induced swimming, how fast the robot can respond to a change of direction in the magnetic field? What are the controlling factors in determining the response speed?

Response: This is another good question. Since the robot's magnetization tend to follow the applied rotating magnetic field in its plane of rotation in millisecond, altering the rotation axis of the magnetic field would immediately change the robot's swimming direction by its rigid body rotation. To address the reviewer's comment, we have tested how fast the robot can change its swimming direction by conducting additional experiments. The robot can turn an angle of 180° in less than 0.2s. This speed is sufficiently fast for the applications we are aiming at. A demonstration video link can be found here:

<https://www.dropbox.com/s/i8cazn9vpd7rpov/Video%20for%20fast%20steering.mp4?dl=0>

Comment #3 - Can the authors comment on the feasibility of extending their approach to other form of origami? Or maybe Kresling is the only one due to its unique shape and folding mechanism?

Response: Thank the reviewer for this question. There are several reasons that we chose to use Kresling origami in this work.

Global symmetry. The Kresling origami is a triangulated cylinder, whose global aspect ratio (height/diameter of the cross-section) can be easily designed to be close to one to promote omnidirectional rolling locomotion like a sphere does. Based on the rotational motion, the robot can generate effective on-ground rolling and flipping around the axes parallel and perpendicular to its longitudinal direction, respectively.

Tilted panel for propulsion. The Kresling origami has tilted triangular panels distributed radially along its longitudinal axis, which function in a way that is analogous to the blades of a propeller to generate effective propulsion for swimming.

Foldability for pumping. Kresling has coupled rotation-displacement property, which means its folding can be enabled by applying a magnetic torque. At the same time, the Kresling used in this paper is designed to be monostable. Upon removing the magnetic field, the robot can recover to its unfolded state, forming a reversible folding/unfolding pumping mechanism for controllable drug release.

The unique features of Kresling make it the perfect candidate for the multifunctional millirobot with the capabilities of providing amphibious locomotion for navigation and pumping for targeted drug delivery. Although Kresling is the origami pattern that directly give us the merits, we do think other origami patterns can be designed with similar or selected geometric features to achieve the desired functionalities.

Comment #4 - For the Kresling pattern used, the cross-section has a hexagonal shape. Can other shape of cross-section be used (for example, pentagon shape)?

Response: This is a very good question. Other shapes of cross-section (i.e., pentagon, octagon, or other polygons) can also be used. However, the selection of the cross-section shape will influence the robot's performance in different aspects.

First, the cross-section shape influences the rolling performance. For example, a square cross-section is less effective for rolling. As the number of polygon sides increases, the cross-section becomes more like a circle, which is beneficial for the rolling performance.

Second, the cross-section shape influences the swimming performance. The number of tilted panels is the same as the number of polygon sides. At the same time, both the shape of tilted panels and the angle between panels and the polygon will change if we use other polygons. This shape and angle are critical for the propulsion generated by the propeller-like structure. These features will result in different propulsion generated by the propeller-like structure.

Third, the cross-section shape influences the folding performance. As can be seen in [R3], with the same diameter of the polygon, the mechanical properties of Kresling patterns with different cross-section shapes are characterized. The peak force needed to fold the Kresling pattern increases with the number of polygon sides.

The design space of the Kresling pattern is very large. The cross-section shape and all features mentioned above (i.e., cross-section geometry, tilted panel geometry, and global aspect ratio) should be taken into account based on the requirements of the specific application. We are also working on the optimization of the Kresling pattern for enhanced swimming performance.

[R3] Nayakanti, N., Tawfick, S. H., & Hart, A. J. (2018). Twist-coupled kirigami cells and mechanisms. *Extreme Mechanics Letters*, 21, 17-24.

Reviewer #2

General Comments:

This article reported a wireless multi-modal origami millirobot capable of on-ground rolling/flipping, obstacle crossing, jumping, spinning-enabled swimming under water and at the interface, with integrated functions, such as medicine delivery and cargo transportation. In this work, the Kresling origami structure is incorporated into the millirobot. Based on the geometrical features of Kresling origami, the self-adaptive rolling-to-flipping on-ground omnidirectional movement can be achieved, when the robot crosses an obstacle (e.g., wall, spaced stairs, and inclined surface). When two magnetic plates are integrated into the robot, the compressive and rotational deformations of Kresling structure can be achieved to enable the release of medicine. Taking advantage of the Kresling's tilted triangular panels distributed radially along its axis, a net propulsion force can be generated when the robot is actuated by magnetic field spins. The demonstrations that the amphibious robot can move in various man-made environments and ex vivo pig stomach with/without viscous fluid were further provided.

Overall, the millirobot reported in this paper is very impressive, with a rich diversity of locomotion modes. The paper is thereby recommended for publication in Nature Communications, after minor revisions noted below.

Response: We would like to thank the reviewer for the positive feedback and for taking the time to carefully review our paper. We have provided point-by-point addressments of the specific comments from the reviewer.

Specific Comments:

Comment #1 - In Fig. 2C, the author shows that the robot can move on an inclined surface of 30°. What is the maximum angle of the inclined surface on which the millirobot could climb?

Response: Thank you for raising this question. The capability of climbing an inclined surface largely depends on the friction coefficient between the robot and the inclined surface. The inclined surface in the paper is made of rigid material, and its surface is smooth and with low friction. The

maximum angle of the inclined surface the robot can climb is about 30° . However, if the friction coefficient increases, the robot can climb an inclined surface with a higher angle. To better answer this question, we conducted additional experiments by having inclined surfaces coated with a silicone material Ecoflex 00-30, which provides larger friction. The maximum angle of the inclined surface on which the robot can climb is 50° , as shown in Fig. R1 and the video. Here, the used magnetic field parameters are the same as the demonstrations in the paper. The angle can be further enhanced by increasing the friction coefficient and the robot's driving force.

Fig. R1. The robot climbs an 50° inclined surface coated with high friction material.

Video: <https://www.dropbox.com/s/8lxsh3d4b84o3xv/video%20for%20climbing%20inclined%20surface.mp4?dl=0>

Comment #2 - In Fig. 3F-I, the swimming performances of the robots along different routes are demonstrated. When the plane of rotated magnetic field is parallel to the axis of Kresling structure in the beginning, can the robot adaptively adjust its swimming mode and eventually move in a spinning mode?

Response: Thank the reviewer for this question. Regardless of the robot's initial orientation, it always adjusts itself to the spinning mode in water under a rotating magnetic field. To see this, if the robot's magnetization is not in the plane of the rotating magnetic field, the rotating magnetic field first generates a torque to align the robot's longitudinal axis with the rotating axis of the applied magnetic field. After that, the magnetization will follow the magnetic field in its rotating plane, and the robot will swim along its longitudinal axis. Therefore, once the plane of the rotating magnetic field is set, the robot always swims along the desired direction in a spinning mode regardless of its initial orientation.

Comment #3 - On Page 3, Line 92 of supplementary information, the Kresling structure can be assembled into robot after experiencing 400-time cyclic compressions, for consistent and monostable mechanical behavior. Is it possible to achieve desired mechanical property of the Kresling structure without cyclic compressions?

Response: Thank the reviewer for bringing up this question. The folding actuation of the Kresling origami requires relatively soft hinges, while the locomotion such as swimming requires rigid origami panels. To achieve this, we assemble the Kresling origami using polypropylene film, a very thin plastic sheet, whose stiffness is ideal for the rigid origami panel. Then, to create the deformable hinges from this relatively stiff material, we cyclically compress the Kresling sample are to soften the hinges and achieve stable and reliable hinge behavior, so the folding performance is repeatable and predictable for controlled drug release.

Although it is technically possible to use soft elastomer as hinges to save the cyclic compression effort, it will then require another stiff material for the thin rigid panel. This would lead to more required study on how to effectively bond the thin rigid panels with the soft material hinges, which significantly increase the fabrication difficulty and reduce the reliability of the samples as there is a higher chance for interfacial failure.

Based on the reviewer's comments, we have added the following contents in the revised SI.

“The Young's modulus is calculated to be 1577.9 MPa based on 0.5% strain, **which can provide relatively large stiffness for Kresling's panels.**”

“Thus, **to soften Kresling's hinges and obtain a consistent mechanical behavior,** we manually compress samples for 400 cycles before performing mechanical characterization and assembling them with other components for magnetic actuation.”

Comment #4 - Please check the typo in Movie S5. ‘Titled’ should be corrected as ‘tilted’.

Response: We would like to thank the reviewer for catching this typo. We have revised Movie S5 and carefully checked all files (main text, SI, and videos) to ensure that there are no similar mistakes.

Reviewers' Comments:

Reviewer #1:

Remarks to the Author:

The authors have adequately addressed my comments and I will be happy to accept this excellent paper.

Reviewer #2:

Remarks to the Author:

All of my previous comments have been very well addressed. I believe that the paper is now ready for publication.

Responses to Reviewers

Review #1: General Comments: The authors have adequately addressed my comments and I will be happy to accept this excellent paper.

Review #2: General Comments: All of my previous comments have been very well addressed. I believe that the paper is now ready for publication.

Response: We would like to thank the reviewer for taking the time to carefully review our paper and helping us improve the paper quality.